# Exotic plants accumulate and share herbivores yet dominate communities via rapid growth

Warwick J. Allen [1,2 ✉], Lauren P. Waller[1,2], Barbara I. P. Barratt [3,4], Ian A. Dickie [1,5] & Jason M. Tylianakis [1,5]

Herbivores may facilitate or impede exotic plant invasion, depending on their direct and indirect interactions with exotic plants relative to co-occurring natives. However, previous studies investigating direct effects have mostly used pairwise native-exotic comparisons with few enemies, reached conflicting conclusions, and largely overlooked indirect interactions such as apparent competition. Here, we ask whether native and exotic plants differ in their interactions with invertebrate herbivores. We manipulate and measure plant-herbivore and plant-soil biota interactions in 160 experimental mesocosm communities to test several invasion hypotheses. We find that compared with natives, exotic plants support higher herbivore diversity and biomass, and experience larger proportional biomass reductions from herbivory, regardless of whether specialist soil biota are present. Yet, exotics consistently dominate community biomass, likely due to their fast growth rates rather than strong potential to exert apparent competition on neighbors. We conclude that polyphagous invertebrate herbivores are unlikely to play significant direct or indirect roles in mediating plant invasions, especially for fast-growing exotic plants.

---

[1] The Bio-Protection Research Centre, School of Biological Sciences, University of Canterbury, Christchurch, New Zealand. [2] The Bio-Protection Research Centre, Lincoln University, Lincoln, New Zealand. [3] AgResearch, Invermay Research Centre, Mosgiel, New Zealand. [4] Department of Botany, University of Otago, Dunedin, New Zealand. [5] These authors jointly supervised this work: Ian A. Dickie, Jason M. Tylianakis. ✉email: warwick.j.allen@gmail.com

An extensive body of research has sought to understand how natural enemies influence the success and impacts of exotic plant species. Despite these efforts, we still lack clarity around how differences in natural enemy preference (i.e., degree of specialisation) and provenance (i.e., native or exotic) can influence invasion success in communities. Several predictions can be derived from the scores of hypotheses and sub-hypotheses that have been proposed[1]. For example, one prediction of the enemy release hypothesis[2,3] with strong support across multiple systems and methodological approaches is that exotic plant species escape from regulation by monophagous (i.e., feeding on a single host plant species) or oligophagous (i.e., feeding on a narrow range of host plant species, often constrained to a single genus or family) natural enemies that were present in their native range (herein 'biogeographical enemy release')[4–8]. How exotic species interact with polyphagous (i.e., feeding on a broad range of host plant species) enemies in the introduced range, however, is less clear. Enemy release theory predicts that exotic species should benefit from weaker interactions with polyphagous enemies relative to co-occurring native species (herein 'community enemy release')[3], whereas the biotic resistance hypothesis predicts that resident polyphagous enemies should inhibit exotic species more than natives[9].

Evidence supporting these contrasting predictions has also been equivocal[1,6,8,10]. For example, some plant invaders are successful because they possess novel defences never before encountered by native herbivores (i.e., the novel weapons hypothesis)[11–13], such as *Alliaria petiolata* (garlic mustard) in North America[14]. Conversely, other invaders are readily incorporated into the diet of the resident herbivore community, such as *Cirsium vulgare* (Scotch thistle) in Nebraska, USA[15]. One research method has focused on comparing important invaders with congeneric native species[16–19], frequently finding support for invader escape from enemies[5]. Despite this being a well-reasoned approach, the majority of studies to date have examined a relatively low diversity of plants and enemies from the community (see Supplementary Table 1 for sample sizes of species from studies used in the Meijer et al. 2016 analysis[7]), leaving it unclear whether plant–herbivore interactions systematically favour exotic species. To this end, several impressive field surveys and common garden experiments have sought to describe more general patterns of herbivore diversity, abundance and damage on multiple native and exotic plants, but also with mixed results[20–27]. For example, field surveys of 47 plant species in Japan and the Netherlands supported the enemy release hypothesis, finding higher insect herbivore diversity, abundance, biomass and damage on native plants than on exotic plants[26]. In contrast, feeding assays involving 57 native and 15 exotic plant species showed the opposite, with native polyphagous crayfishes preferring exotic plants[22]. Furthermore, observed enemy diversity, density and damage does not always translate into proportional reductions in plant fitness (i.e., biomass, flowering, seed production) by herbivores[28]. This may be especially true for exotic plant species, which can mitigate herbivore impacts via typically fast growth rates[29] and high tolerance of herbivory[30]. Despite being crucial to understanding the complex interactions in communities of native and exotic plants and natural enemies, multispecies studies that experimentally manipulate enemies and link their damage to performance of native and exotic species have been rare (Supplementary Table 1).

With the arrival of exotic species showing no sign of abating[31], how exotic enemies integrate into novel communities has also received increased research attention. For example, the enemy of my enemy hypothesis posits that co-introduced enemies should cause greater harm to native than exotic species, based on the potential lack of co-evolved defenses[5,32]. Alternatively, exotic herbivores could cause greater harm to exotic than native species if native plants possess defences that are novel to exotic herbivores[11]. However, there is growing evidence that native plants suffer strong impacts from polyphagous exotic herbivores and generalist pathogens[22,23,33], and exotic plants may likewise suffer disproportionate attack from native enemies (i.e., biotic resistance[9,23]). Hence, including both native and exotic enemies in studies of plant–herbivore interactions is important for understanding how polyphagous herbivores influence plant invasions.

Indirect species interactions are of growing interest to invasion ecologists[34,35]. For example, apparent competition (i.e., negative interactions between two or more species mediated by changes in the population or behaviour of shared natural enemies[36,37]) can influence the ability of species to invade (i.e., indirect biotic resistance) and their impacts on the community[38–40]. Moreover, as exotic species accumulate both species richness and biomass of enemies over time, the potential for apparent competition (PAC) with other native and exotic species is likely to increase. If the enemy is also exotic, and native competitors are disproportionately impacted, this would represent an indirect invasional meltdown (i.e., facilitation between two or more exotic species[41]), with potential implications for management strategies[35]. However, because experimental tests of how natural enemies interact with native and exotic species are usually conducted in isolation from other species in the community, comparisons of indirect interactions between native and exotic taxa are lacking in the literature, apart from two examples that we are aware of[40,42]. Moreover, recent evidence suggests that community-level outcomes of apparent competition can be predicted with some success based on the sharing of interaction partners[43–45], but this approach has yet to be attempted in an invasion context or outside of host–parasitoid trophic systems.

Another indirect interaction of interest to ecologists is the effect of soil biota (e.g., bacteria, fungi, nematodes and other microorganisms) on herbivores via changes in host plant nutritional quality, defenses and other plant traits[46,47]. As with herbivores, soil biota could have variable impacts on the success of exotic species, depending upon their degree of specialisation and the relative influence of harmful and beneficial taxa. Plant–soil feedback experiments aim to quantify the interplay between plant species and their associated soil communities[48,49] and represent one way to test the impacts of soil biota on plants. For example, the effect of specialist soil biota can be estimated by comparing plant performance in soil conditioned by conspecifics (i.e., 'home' soils meant to mimic established invasions where specialist soil biota are present) and heterospecifics (i.e., 'away' soils meant to mimic uninvaded communities where specialist soil biota are absent). There is mixed evidence regarding whether the presence or absence of specialist soil biota should favour exotic plant species and lead to invasions[50,51]. However, multiple meta-analyses have indicated that relative to native species, exotic plants may perform better in their own 'home' soils than 'away' soils from other species[49,51,52], suggesting that specialist soil biota disproportionately benefit exotic plants once they have established. Whether the subsequent indirect impacts of specialist soil biota on herbivores counteract these benefits remains untested, although the plant vigour hypothesis predicts that plants that benefit more from soil biota may also experience stronger herbivory[53,54]. To our knowledge, no studies to date have explored these questions in plant communities with simultaneous manipulation of plant–herbivore and plant–soil biota interactions.

Here, we use a large-scale experiment to examine the direct and indirect interactions of exotic and native plants with a mix of common oligophagous and polyphagous native and exotic

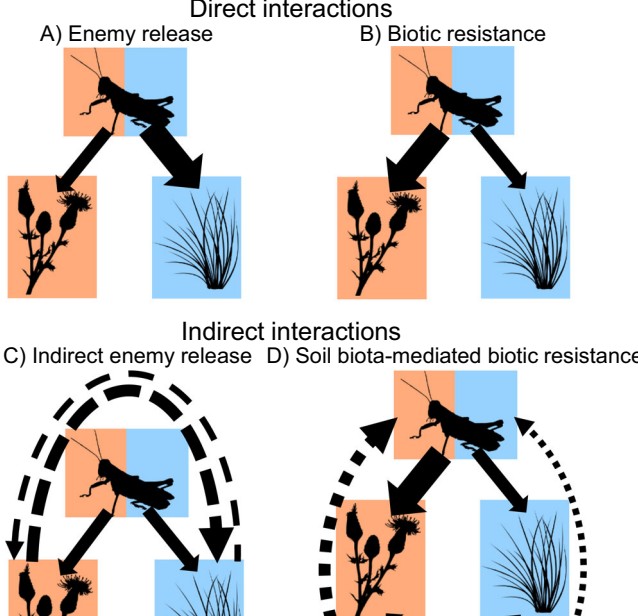

## Direct interactions
### A) Enemy release
### B) Biotic resistance

## Indirect interactions
### C) Indirect enemy release   D) Soil biota-mediated biotic resistance

**Fig. 1 Conceptual figure detailing the invasion hypotheses tested. A** The enemy release hypothesis, predicting that exotic plants should benefit from weaker interactions with polyphagous herbivores relative to co-occurring native species; **B** the biotic resistance hypothesis, predicting that resident polyphagous herbivores should inhibit exotic plants more than natives; **C** indirect enemy release, predicting that exotic plants should have higher potential to exert and lower potential to receive apparent competition than native plants; and **D** soil biota-mediated biotic resistance, predicting that exotic plants should experience stronger interactions with herbivores when growing in soil communities containing specialist soil biota. See main text and Table 1 for additional details on hypotheses and predictions. Arrow width represents the relative strength of negative direct (solid line), negative indirect (dashed line) and positive indirect (dotted line) interactions, and panel colour represents native (blue), exotic (orange) or mixed (both colours) provenance of plants, herbivores and soil biota. Symbols courtesy of the Integration and Application Network (ian.umces. edu/symbols/).

herbivores. We manipulate and measure plant–herbivore and plant–soil biota interactions in 160 mesocosm grassland communities, designed from a pool of 39 plant and 20 invertebrate herbivore species that varied in provenance, phylogeny and traits[55]. We integrate several invasion ecology hypotheses to address four overarching research questions: (1) Compared with native plant species and native-dominated communities, do exotic plant species and exotic-dominated communities experience weaker (i.e., enemy release; Fig. 1A) or stronger (i.e., biotic resistance; Fig. 1B) interactions (measured as herbivore diversity, biomass and damage) with native and exotic herbivores? (2) Do exotic plants experience lower or higher proportional reductions in biomass production from herbivores than native plants (i.e., enemy release or biotic resistance translate into impacts on plant fitness; Fig. 1A, B, respectively)? (3) Do exotic plants have higher potential to exert and lower potential to receive apparent competition than native plants, with consequences for plant biomass (i.e., indirect enemy release; Fig. 1C)? (4) Do exotic plants experience stronger interactions with herbivores when growing in soil communities containing specialist soil biota (i.e., soil biota-

mediated biotic resistance against established invasions; Fig. 1D)? These research questions also comprise multiple specific predictions that are outlined in Table 1 and the Methods section. We find that compared with natives, exotic plants support higher diversity and biomass of native and exotic herbivores, and experience larger proportional biomass reductions from herbivory, regardless of whether specialist soil biota are present. Yet, exotic plants dominate community biomass, likely via their fast growth rates rather than apparent competition with neighbours. We conclude that polyphagous invertebrate herbivores are unlikely to play significant direct or indirect roles in mediating invasions of fast-growing exotic plants.

## Results

**Exotic plant species and exotic-dominated communities experienced stronger interactions with native and exotic herbivores.** Regardless of herbivore provenance or soil treatment, herbivore species were more than twice as likely to interact with exotic than native plants (plant provenance: $F = 5.93$, $P = 0.015$; Fig. 2A and Supplementary Table 2) and achieved 72% higher biomass on exotics than natives ($F_{1,41} = 24.71$, $P = 0.00001$; Fig. 2B and Supplementary Table 3). Exotic herbivore biomass per mesocosm increased with the proportion of exotic species planted (slope = 1.78, $t = 4.29$, $P = 0.00009$; Fig. 2C), while no relationship was observed for native herbivore biomass (slope = 0.28, $t = 0.68$, $P = 0.501$; plant provenance × herbivore provenance interaction: $F_{1,134} = 43.67$, $P = 8.4e^{-10}$; Supplementary Table 4). Although high herbivore biomass could amount to proportionally low herbivore biomass for plants with high biomass (i.e., promoting enemy release), the herbivore biomass to plant biomass ratio did not differ between native and exotic plants ($F_{1,38} = 1.35$, $P = 0.253$; Supplementary Table 5). Mirroring the result for herbivore biomass, the relationship between mesocosm herbivore:plant biomass ratio and the proportion of exotic species planted depended upon herbivore provenance ($F_{1,76} = 37.86$, $P = 3.3e^{-8}$; Supplementary Table 6), increasing for exotic herbivores (slope = 1.90, $t = 3.94$, $P = 4.4e^{-11}$) but not for native herbivores (slope = 0.40, $t = 0.83$, $P = 0.409$).

Plants interacted with just over half (56 ± 1%, mean ± SEM) of the herbivore species in their mesocosm. Plant normalised degree (i.e., the proportion of herbivore species that fed upon a given host plant out of the total herbivore species in the mesocosm) did not differ between native and exotic plants ($F_{1,48} = 1.35$, $P = 0.251$; Fig. 2D and Supplementary Table 7), although herbivore species richness of mesocosms increased with the proportion of exotic species planted in the community (slope = 0.41, $F_{1,18} = 9.65$, $P = 0.002$; Fig. 2E and Supplementary Table 8).

Herbivore chewing and scraping damage to plants was low throughout the experiment, with an average of 4.3 ± 0.2% of leaf tissue removed across all plants in +Herbivore mesocosms. Average damage to exotic plants was almost double that of native plant species, although this effect was non-significant ($F = 12.76$, $P = 0.062$; Fig. 2F and Supplementary Table 9). A similar result was observed at the mesocosm level, where mean herbivore damage per plant did not vary with the proportion of exotic species planted (slope = −0.42, $F = 6.53$, $P = 0.116$; Supplementary Table 10).

**Exotic plants experienced higher proportional reductions in biomass in mesocosms with herbivores, yet still dominated plant community biomass.** Exotic plants produced 31% less total biomass in +Herbivore compared with −Herbivore mesocosms ($P = 0.012$, Bonferroni corrected pairwise Tukey test based on the plant provenance × herbivore treatment interaction: $F_{1,884} = 4.08$, $P = 0.044$; Fig. 3A and Supplementary Table 11), whereas the

**Table 1 Overarching research questions and specific predictions tested using our mesocosm experiment.**

| Category | Overarching research question (numbered) and specific predictions (lowercase letters) |
|---|---|
| Direct plant–herbivore interactions | 1. Compared with native plant species and native-dominated communities, do exotic plant species and exotic-dominated communities experience weaker (i.e., enemy release; Fig. 1A) or stronger (i.e., biotic resistance; Fig. 1B) interactions (measured as herbivore diversity, biomass and damage) with native and exotic herbivores?<br>(a) Compared with native plants, exotic plants and exotic-dominated communities accumulate less native herbivore biomass (both total and proportional to plant biomass) and more exotic herbivore biomass.<br>(b) Exotic plants and exotic-dominated communities host fewer herbivore species than natives.<br>(c) Exotic plants and exotic-dominated communities suffer lower herbivore damage than natives. |
| Net herbivore impact on plant biomass and exotic dominance | 2. Do exotic plants experience lower or higher proportional reductions in biomass production from herbivores than native plants (i.e., enemy release or biotic resistance translate into impacts on plant fitness; Figs. 1A, B)?<br>(a) Exotic plants experience lower proportional reductions in total, belowground and aboveground biomass production from herbivores compared to natives.<br>(b) Exotic plants make up a disproportionate proportion of plant community biomass, especially when herbivores are present. |
| Indirect plant–herbivore interactions | 3. Do exotic plants have higher potential to exert and lower potential to receive apparent competition than native plants, with consequences for plant biomass (i.e., indirect enemy release; Fig. 1C)?<br>(a) Exotic plants have higher $PAC_{exerted}$ and lower $PAC_{received}$ than natives.<br>(b) Plants with higher $PAC_{received}$ have less biomass and more herbivore damage.<br>(c) Plants with more biomass exert higher $PAC_{exerted}$. |
| Indirect soil biota–plant–herbivore interactions | 4. Do exotic plants experience stronger interactions with herbivores when growing in soil communities containing specialist soil biota (i.e., soil biota-mediated biotic resistance against established invasions; Fig. 1D)?<br>(a) Exotic plant biomass will increase and native plant biomass decrease in soils containing specialist soil biota (i.e., 'home' soil).<br>(b) Community enemy release of exotic plants from herbivores will be reduced in soils containing specialist soil biota (i.e., 'home' soils). |

herbivore treatment did not affect native plant total biomass ($P = 1.000$). Exotic plants had 3.8 and 5.7 times higher total biomass than native plants in +Herbivore and −Herbivore mesocosms (Fig. 3A), respectively, but these effects were non-significant due to high variability in plant biomass ($P = 0.235$ and $0.084$, respectively). To examine if reduced biomass production of plants due to herbivores differed below and aboveground, we repeated the analysis for these separate biomass partitions. For belowground biomass, the results were similar to those of total biomass (plant provenance × herbivore treatment interaction: $F_{1,883} = 8.25$, $P = 0.004$; Supplementary Fig. 1 and Supplementary Table 12), except that exotic plants produced seven times more belowground biomass than natives, but only when herbivores were absent ($P = 0.029$). Exotic plants had 5.8 times higher aboveground biomass than natives ($F_{1,36} = 5.52$, $P = 0.024$; Supplementary Fig. 2A and Supplementary Table 13), regardless of the herbivore treatment (plant provenance × herbivore treatment interaction: $F_{1,888} = 3.69$, $P = 0.055$; Supplementary Table 13), and the 20% reduction in biomass production due to the herbivore treatment was consistent for native and exotic plants ($F_{1,884} = 7.56$, $P = 0.006$; Supplementary Fig. 2B and Supplementary Table 13).

For total plant biomass of mesocosms, there was a significant interaction between the proportion of exotic plants and the soil treatment ($F_{1,134} = 4.27$, $P = 0.041$; Supplementary Table 14), although plant biomass did not vary with the proportion of exotics planted for either soil treatment ('home': slope $= -0.38$, $t = -1.29$, $P = 0.198$; 'away': slope $= -0.12$, $t = -0.53$, $P = 0.600$; Supplementary Fig. 3A). Belowground plant biomass decreased with the proportion of exotics planted in the 'home' soil treatment (slope $= -1.21$, $t = -2.44$, $P = 0.021$) but not in 'away' soil (slope $= -0.65$, $t = -1.31$, $P = 0.200$; proportion of exotic plants × soil treatment interaction: $F_{1,134} = 4.93$, $P = 0.028$;

Supplementary Fig. 3B and Supplementary Table 15). Aboveground biomass did not vary with the proportion of exotic plants, herbivore and soil treatments, or any interactions among them (all $P > 0.056$; Supplementary Fig. 3C and Supplementary Table 16).

Finally, exotic plants dominated communities that they were planted into, consistently making up a significantly greater proportion of the mesocosm biomass than expected (Fig. 3B). Moreover, because 95% confidence intervals overlapped between levels of the herbivore treatment (Fig. 3B), herbivory did not appear to significantly alter exotic plant dominance.

**Exotic plants had higher potential to exert but not receive apparent competition than did native plants.** Exotic plants generated 14 times higher $PAC_{exerted}$ than did native plant species ($F_{1,40} = 7.07$, $P = 0.011$; Fig. 4A and Supplementary Table 17), whereas the 68% higher average $PAC_{received}$ observed for exotic than native plants was non-significant ($F_{1,38} = 0.07$, $P = 0.575$; Fig. 4B and Supplementary Table 18). In other words, exotic plants tended to share herbivores with many other species, and potentially shared them more with other exotics, though this was highly variable. The soil treatment and its interactions did not influence PAC (all $P > 0.188$; Supplementary Tables 17 and 18).

We also explored the causes and consequences of PAC, finding that plants with more biomass exhibited stronger potential to exert apparent competition on the community (slope $= 0.004$, $F_{1,437} = 23.74$, $P = 0.000002$; Supplementary Fig. 4), and plants that experienced higher $PAC_{received}$ also had lower biomass (slope $= -0.0007$, $F_{1,899} = 5.26$, $P = 0.022$; Supplementary Fig. 5). However, the latter relationship did not vary between the added and reduced herbivore treatments (plant biomass × herbivore treatment interaction: $F_{1,892} = 3.45$, $P = 0.064$), indicating that

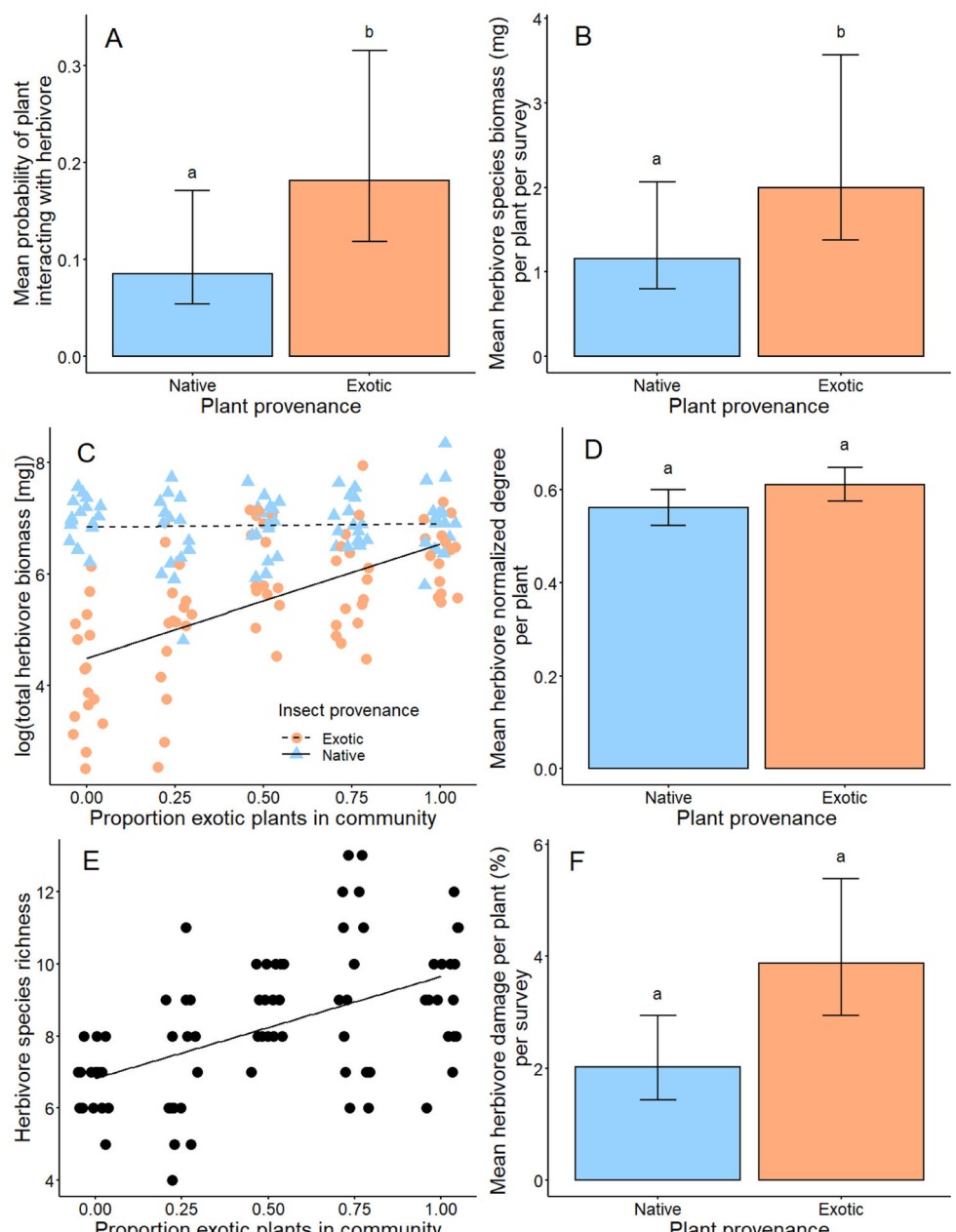

**Fig. 2 Plant–herbivore interactions of native and exotic plants and their composite communities in the mesocosm experiment. A** Herbivore species were twice as likely to interact with exotic (orange, $n = 2876$ potential plant–herbivore interactions, with 961 realised) than native (blue, $n = 2652$ potential plant–herbivore interactions, with 646 realised) plant species within their fundamental host range ($F = 5.93$, $P = 0.015$). **B** Mean herbivore species biomass was 72% higher on exotic ($n = 1333$ plant–herbivore interactions) than native ($n = 809$ plant–herbivore interactions) plant species ($F_{1,41} = 24.71$, $P = 0.0001$). **C** Exotic herbivore total biomass per mesocosm (log-transformed; orange circles) increased with the proportion of exotic species planted into mesocosm communities (slope $= 1.78$, $t = 4.29$, $P = 0.00009$), whereas no relationship was observed for native herbivores (blue triangles, slope $= 0.28$, $t = 0.68$, $P = 0.501$; plant provenance × herbivore provenance interaction: $F_{1,134} = 43.67$, $P = 8.4e^{-10}$; $n = 80$ mesocosms per herbivore provenance). **D** Mean herbivore species richness (quantified as normalised degree, the proportion of interactions observed out of all possible interactions) did not differ between native ($n = 193$) and exotic ($n = 242$) plants ($F_{1,48} = 1.35$, $P = 0.251$). **E** Herbivore species richness of mesocosm communities ($n = 80$) increased with the proportion of exotic plant species planted (slope $= 0.41$, $F_{1,18} = 9.65$, $P = 0.002$). **F** Mean percent chewing and scraping damage to leaf tissue from invertebrate herbivores did not significantly differ between native ($n = 320$) and exotic ($n = 320$) plants ($F = 12.76$, $P = 0.062$). Different lowercase letters indicate significant differences ($P < 0.05$) between back-transformed estimated marginal means (±SEM) from (generalised) linear mixed models. Scatterplot linetypes indicate slopes that significantly differ from zero (solid lines, $P < 0.05$) or do not (dashed lines). A small amount of jitter has been added to separate overlapping points on the x-axis. Corresponding violin plots showing the distribution of raw data are presented in Supplementary Fig. 15.

the relationship was likely driven by the direct effects of increased herbivore abundance in +Herbivore mesocosms, rather than indirect interactions mediated by herbivores. Herbivore chewing and scraping damage did not increase with $PAC_{received}$ (slope $= 0.0002$, $F = 1.65$, $P = 0.206$).

**Plant–soil feedbacks had no impact on plant–herbivore interactions**. The plant–soil feedback soil treatment had little influence on any of the response variables, except for moderating the relationship between proportion of exotic plants and total and belowground plant biomass as described above. The soil

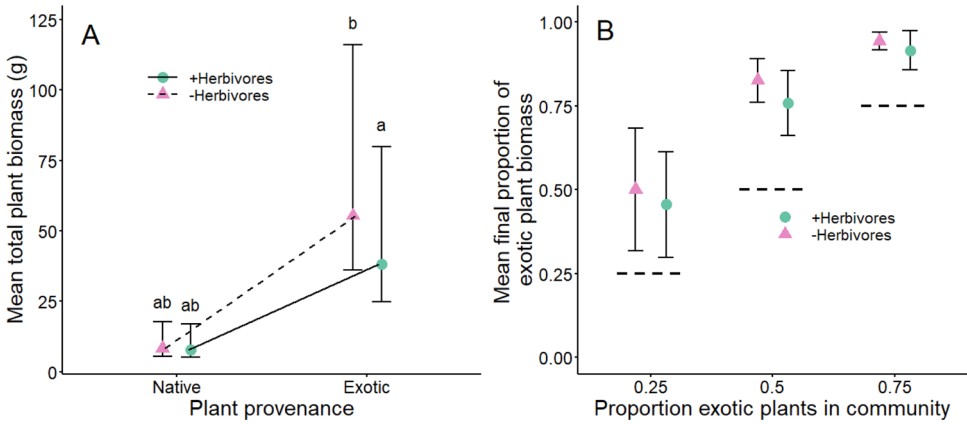

**Fig. 3 Influence of the herbivore treatment on total plant biomass and the proportion of mesocosm biomass made up of exotic plants. A** Exotic plants ($n = 211$ and 206 in −Herbivore and +Herbivore mesocosms, respectively) produced 31% less total biomass in +Herbivore (green circles and solid line) compared with −Herbivore (pink triangles and dashed line) mesocosms ($P = 0.012$, Bonferroni corrected pairwise Tukey test based on the plant provenance × herbivore treatment interaction: $F_{1,884} = 4.08$, $P = 0.044$), whereas the herbivore treatment did not affect native plants ($P = 1.000$, $n = 273$ and 261 in −Herbivore and +Herbivore mesocosms, respectively). Different lowercase letters indicate significant differences ($P < 0.05$, based on Bonferroni corrected Tukey tests) between back-transformed estimated marginal means (±SEM) from linear mixed models. **B** The mean (±95% confidence intervals, $n = 16$) proportion of total mesocosm biomass that was made up of exotic plants was significantly higher than the expected proportion of exotic plant biomass (dashed lines, based on proportion of exotics planted in the community), regardless of herbivore treatment (pink triangles = −Herbivores; green circles = +Herbivores). Any 95% confidence intervals that do not overlap the respective dashed line are considered to be signficiantly different from the expected proportion of exotic biomass. Corresponding violin plots showing the distribution of raw data are presented in Supplementary Fig. 16.

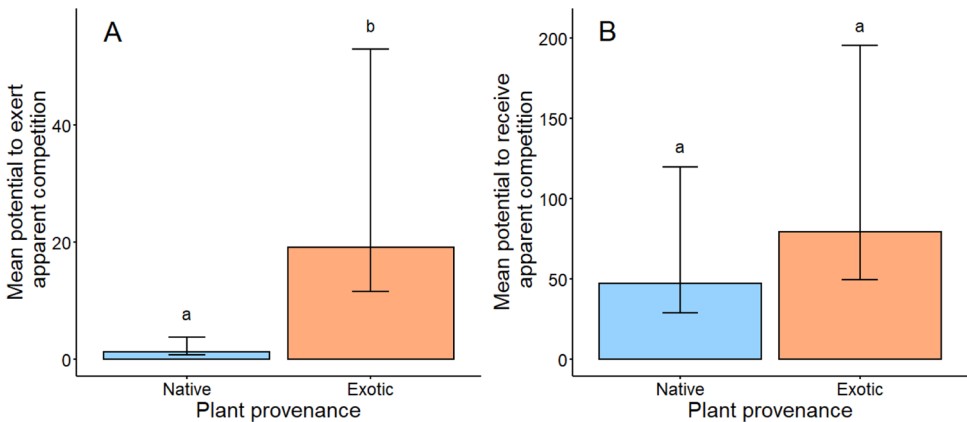

**Fig. 4 Potential for apparent competition (PAC) of native and exotic plants. A** Exotic plant species (orange, $n = 320$) generated 14 times higher PAC$_{exerted}$ compared with native plant species (blue, $n = 320$; $F_{1,40} = 7.07$, $P = 0.011$). **B** PAC$_{received}$ did not significantly differ between native and exotic plants ($n = 320$ per plant provenance; $F_{1,38} = 0.07$, $P = 0.575$). Different lowercase letters indicate significant differences ($P < 0.05$) between back-transformed estimated marginal means (±SEM) from linear mixed models. Corresponding violin plots showing the distribution of raw data are presented in Supplementary Fig. 17.

treatment and its interactions did not influence herbivore presence, richness, biomass, chewing and scraping damage, or the herbivore:plant biomass ratio for either individual plants or mesocosms (all main effects and interactions: $P > 0.091$; Supplementary Tables 2–10).

## Discussion
We found no evidence that exotic plant species experienced weaker interactions with native or exotic oligophagous and polyphagous invertebrate herbivores relative to co-occurring native plant species, contrary to predictions of invasion theory (i.e., 'community enemy release'). Instead, exotic plants and communities accumulated more herbivore species and biomass, resulting in reduced biomass of exotic plants when compared to their counterparts in mesocosms without added herbivores. However, despite suffering higher herbivore richness, biomass

and proportional reductions in biomass production from herbivores, the exotic plants were able to overcome these high levels of herbivory and still dominate the biomass of mesocosm communities in which they occurred. Many herbivore species that attacked exotic plants were also shared with other native and exotic species, indicating that polyphagous herbivores could potentially facilitate exotic plant success by mediating indirect impacts on the surrounding community. Yet, despite this potential, we found little evidence that these indirect interactions influenced herbivore chewing and scraping damage or plant biomass. By incorporating indirect interactions (i.e., PAC) into enemy release theory, assessing high taxonomic and functional diversity of plant–herbivore interactions (i.e., 39 plant and 20 herbivore species), and quantifying multiple measures of enemy release (i.e., herbivore richness, biomass, damage and proportional reductions in biomass production from herbivores), our study represents one of the most comprehensive tests of

community enemy release and biotic resistance to date (see Supplementary Table 1 for comparison to studies cited in the Meijer et al. 2016 meta-analysis[7]).

In contrast to our predictions of 'community enemy release' of exotic relative to native plants[3], we found that exotic plants and exotic-dominated communities consistently suffered higher total herbivore richness, biomass and proportional reductions in biomass production from herbivores than native plants. However, the herbivore to plant biomass ratio did not differ between native and exotic plants, indicating that plants with higher biomass may simply accumulate higher species richness and biomass of polyphagous herbivores. Furthermore, the higher herbivore loads on exotic plants reduced their biomass by over 30%, whereas herbivory had no impact on native plant biomass, the complete opposite of our prediction and supporting mild biotic resistance instead of community enemy release in this experiment. Average damage to plant tissue from chewing and scraping herbivores was only 4.3% of leaf tissue removed, and this did not differ between native and exotic plants. However, this level of damage is similar to the average of 7.5% observed across the plant Kingdom[56], and herbivore damage can translate to variable impacts on plant fitness, from complete defoliation and death through to tolerance and overcompensatory growth. Moreover, the impact of the herbivore treatment on plant biomass production only differed between native and exotic plants for belowground biomass, suggesting that exotic plants either altered their biomass allocation to compensate for aboveground damage or suffered disproportionate impacts of belowground herbivores, primarily from the native New Zealand grass grub (*Costelytra giveni*). Thus, we consider the sevenfold larger effect of the herbivore treatment on biomass production compared to chewing and scraping damage to be a more direct measure of the net impact of herbivores (i.e., the sum of biomass lost from chewing herbivores, unquantified damage from sucking insects and belowground herbivores and reduced growth of impacted plants). Finally, given that exotic but not native plants experienced reduced biomass production in mesocosms with added herbivores, we conclude that the native plants in our experiment may be more tolerant of herbivory than exotic plants, supporting the findings of some studies[57,58] but not others[30,59].

Despite the strong reductions in biomass production due to polyphagous resident herbivores, exotic plants tended to dominate communities into which they were planted. Their dominance may be driven by faster growth rates relative to native species, as they invest in fast-growing and short-lived leaf tissue, characterised by their higher specific leaf area (SLA) than native plants[55]. However, 'biogeographical enemy release' could still play a role, whereby exotic plants escape from monophagous and oligophagous natural enemies present in their native range[3]. This type of enemy release has strong empirical support[4–8], including for several of the exotic plant species in our experiment that have been managed with varying degrees of success using exotic biological control agents[60,61]. Moreover, escape from monophagous and oligophagous enemies could lead to selection for plants that have reduced investement in plant defenses and increased growth and competitive ability (the 'evolution of increased competitive ability [EICA] hypothesis'[62]). Support for the EICA hypothesis has been mixed[63,64], including for species used in our experiment[65], although we did not directly test its predictions in our experiment. Finally, fast growth is characteristic of many invasive plants around the world[29], suggesting that our findings may be generalisable across plant invasions.

Our results oppose those of a large-scale field survey that found lower insect herbivore richness, abundance, biomass and damage on 19 exotic plants compared with 19 native plants[26], indicating that findings may differ between controlled experiments and in the field. There are several potential explanations for these conflicting results. For instance, our experiment has the key advantage of manipulating herbivore presence, allowing us to overcome the lack of consistent translation of herbivore load and damage to proportional reductions in plant fitness[30]. We must also acknowledge that our mesocosm communities did not replicate natural communities, which are almost certainly affected by greater herbivore diversity, indirect effects of natural enemies (i.e., predators and parasitoids)[66,67] and herbivore aggregation, heterogeneity and neighbourhood effects over larger spatial scales[68]. Thus, our results should be taken with caution when translating to natural systems. Moreover, interactions with other antagonists (e.g., pathogens and competitors) and mutualists (e.g., mycorrhiza, rhizobia and endophytes) can also differ between native and exotic plants[4,69,70] and alter plant–herbivore interactions[47,71]. These unmeasured indirect interactions mean that exotic plants may dominate communities via other unexplored mechanisms, such as escape from pathogens, stronger interactions with mutualists (i.e., the 'enhanced mutualism hypothesis'[72]) or stronger competitive ability[73]. Therefore, we suggest that future research considers taking a whole-systems species interaction network approach towards understanding the causes and consequences of biological invasions in novel communities[74]. In practice, this may involve studies that manipulate and integrate multiple different interaction types and examine the consequences for community productivity and function[75,76].

The native and exotic plant species grown in this experiment were representative of those that occur in the New Zealand landscapes from where soil inoculum was collected. However, successful exotic plants differ from natives in several key traits, which could also have influenced the results. For example, legumes are much more common among the New Zealand exotic flora (>100 naturalised species) than the native flora (four genera with ~34 species)[77], which meant that we included six exotic legume species (Fabaceae) and just one native legume in the experiment. Because they often have highly palatable leaves, the disproportionate number of legume species had the potential to increase overall herbivory on exotics, although we only observed strong plant–herbivore interactions for one exotic legume species, *Lupinus arboreus*. Similarly, exotic plant species in this system tended to be fast-growing species adapted to disturbed habitats, whereas many native species favoured a more conservative growth strategy, and therefore may have also invested more in plant defences, which we did not measure directly. To further understand how traits may have mediated differences in plant–herbivore interactions (i.e., herbivore presence, biomass, diversity and damage to plants) between native and exotic plant species, we quantified whether variation in these response variables could be explained by the main effects and interactions of plant provenance with several traits of plants and herbivores (see Supplementary Notes for details on these analyses). However, we found no consistent relationships between traits and plant–herbivore interactions, with results depending upon the response variable and trait being investigated (see Supplementary Tables 19–22 and Supplementary Figs. 6–11 for detailed results). Furthermore, because plant–enemy interactions may not differ between native and naturalised but non-invasive plant species[27,78,79], one could question whether the high herbivory observed on exotic plants was because they were non-invasive species. However, 90% of the exotic plants used for our experiment are considered invasive weeds with ecological and economic impacts[80,81]. Yet, these exotic plants still experienced stronger interactions with resident herbivores compared to native plants, which we suggest makes our results even more surprising and divergent with expectations based on invasion theory[2,3].

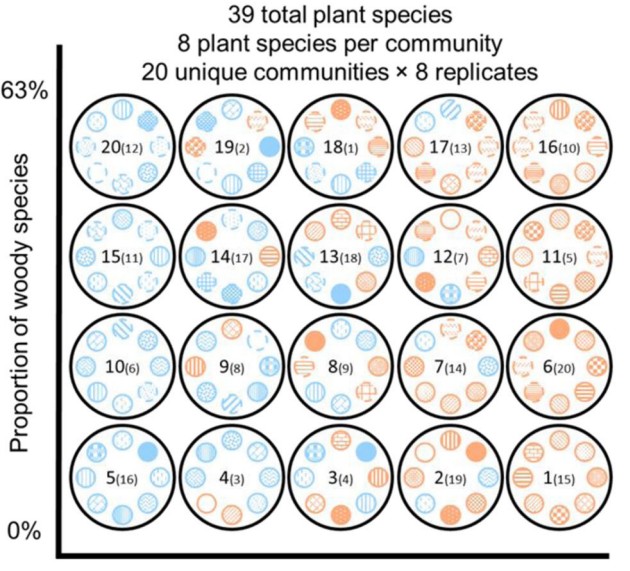

## (A) Experimental design

39 total plant species
8 plant species per community
20 unique communities × 8 replicates

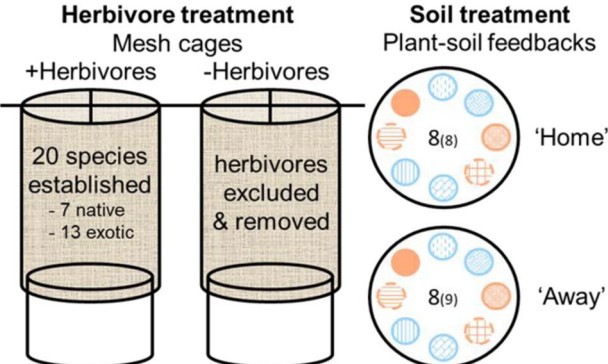

## (B) Data collection & analyses

**8 herbivore surveys:**

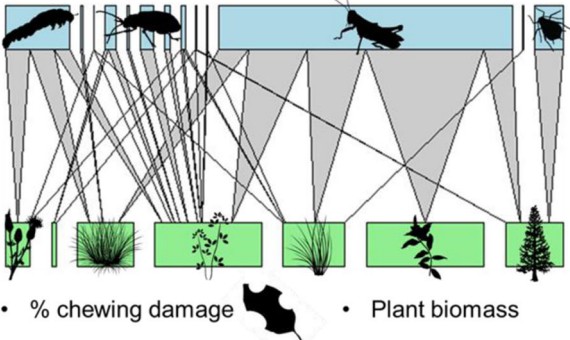

- Biomass ($\alpha$) of each herbivore species ($k$ of $l$ shared species) on each plant pair ($i, j$ of $m$ species)

- % chewing damage    • Plant biomass

- Potential for apparent competition (PAC):
$$d_{ij} = \sum_{k=1}^{p} \left[ \frac{\alpha_{ik}}{\sum_{l=1}^{p} \alpha_{il}} \frac{\alpha_{jk}}{\sum_{m=1}^{H} \alpha_{mk}} \right]$$

**Fig. 5 Conceptual figure detailing the experimental design, data collection and analyses.** The upper portion of the figure shows the experimental design (**A**), with orthogonal exotic and woody gradients of the 20 unique plant communities (numbered, with the community used for the 'away' soil treatment in parentheses), where plant provenance, functional group and plant species are represented by symbol colour (blue = native plant species, orange = exotic plant species), outline (solid = herbaceous, dashed = woody), and pattern (key in Supplementary Table 24), respectively. Also shown are details of the herbivore (mesh cages with herbivore addition and exclusion) and soil (plant–soil feedback; 'home' = soil from conspecifics and 'away' = soil from heterospecifics) treatments; The lower portion of the figure details the data collected for analyses (**B**), including plant and herbivore biomass, herbivore diversity, herbivore chewing and scraping damage, and the equation for pairwise potential for apparent competition[86]. Symbols courtesy of the Integration and Application Network (ian.umces.edu/symbols/).

of a meta-analysis that demonstrated suppression of evolutionarily naïve exotic plants by native polyphagous herbivores, whereas exotic polyphagous herbivores reduced native plant abundance and helped to facilitate exotic plants[23]. One possible reason for our contrasting findings could again be the differences between herbivore exclusion field experiments and our mesocosm experiment. For example, we focused exclusively on invertebrate herbivores, whereas the main subjects of the studies analysed by Parker et al. (2006)[23] were mostly vertebrate polyphagous herbivores. Regardless, the contradictory results further highlight how the outcomes of plant–herbivore interactions involving exotic species may depend upon several factors, including plant provenance, herbivore type, time since introduction, coevolutionary history, plant invasion status (i.e., naturalised vs. invasive), herbivore host plant range and plant and herbivore traits[27].

In partial support of our predictions, exotic plants demonstrated stronger potential to exert apparent competition on other plants, supporting the idea that exotic plants may dominate communities through spillover of accumulated herbivores onto co-occurring species[38,40]. Nevertheless, we found no evidence that these indirect interactions increased herbivore damage or reduced plant biomass on co-occuring plants, or that native plants received stronger indirect impacts than exotics, providing little support for the importance of these indirect interactions or an invasional meltdown. However, our experiment did not assess longer-term indirect impacts, which may be expected to accumulate as invasion progresses[35].

Finally, our soil treatment (i.e., 'home' and 'away' soils meant to mimic communities where soil biota that specialise on the plants are present or absent, respectively) had no indirect impact on any aspect of herbivory, contrasting with our predictions and recent evidence that plant–soil feedback treatments can alter plant–herbivore interactions[82,83]. This result potentially indicates that plant–soil feedbacks involving specialist or generalist soil biota may have little indirect influence on polyphagous invertebrate herbivores at the community level. However, future studies should aim to further investigate why the indirect impacts of soil biota on herbivores observed for individual plants do not translate to the community level. Further, although we did not observe any effects of our soil treatment on herbivores, this does not mean that herbivores did not affect soil biota community composition[84] or their impacts on plant communities[85], but that was not tested here. Moreover, it is clear that different enemy guilds (e.g., herbivores, pathogens) can indirectly impact each other[71], and therefore it may be useful to explicitly incorporate their direct and indirect interactions into studies of plant invasions.

Contrary to our predictions and the 'enemy of my enemy' hypothesis (that exotic enemies should cause less harm to exotic than native species[5,32]), herbivores exhibited over three times higher biomass on exotic than native plants (and no difference in herbivore:plant biomass ratio), regardless of whether they were native or exotic themselves. This result also contrasts with those

In conclusion, we have integrated several invasion hypotheses to investigate the multiple direct and indirect roles that oligophagous and polyphagous herbivores may play in plant invasions. We found exotic plants to be less resistant to and tolerant of oligophagous and polyphagous herbivores than native plants, regardless of herbivore provenance. These findings show that the advantage exotic plant species receive relative to coexisting native species is not congruent with predictions from enemy release theory, and that weaker interactions with resident oligophagous and polyphagous herbivores are unlikely to be a key mechanism driving exotic plant invasions of New Zealand grasslands. Instead, exotic plants experienced mild biotic resistance from herbivores, but ultimately dominated the biomass of plant communities via fast growth. Finally, although the accumulation of herbivores on exotic plants indicated the potential for strong indirect impacts on co-occurring plants, we found no evidence to suggest that this potential was realised through reduced biomass production. Taken together, our findings suggest that oligophagous and polyphagous invertebrate herbivores in the introduced range are unlikely to play a significant role in mediating plant invasions, particularly for fast-growing exotic plants that can compensate for high levels of herbivory.

## Methods

**Experimental design**. We established 160 experimental mesocosm communities (Supplementary Fig. 12), where interactions between plants, invertebrate herbivores and soil biota were manipulated and measured. A previous paper reports the ecosystem level outcomes for the same mesocosms[55], but this paper is the first to explore individual plant–herbivore interactions. Each mesocosm consisted of a 125 L steel pot, with a bottom layer of 22 L of gravel to aid drainage out of the open bottom, 88 L of pasteurised soil and sand (50:50 mixture) and a top layer of 12 L of soil inoculum (see soil treatment details below). Mesocosms were planted with one of 20 unique communities, each consisting of eight plant species (Supplementary Table 23) selected from a pool of 39 plant species that co-occur in New Zealand grassland communities (19 natives, 20 exotics, Supplementary Table 24). Plant species were selected based on their occurrence at sites where inoculum soil was collected, and communities were designed to vary orthogonally in their proportion of exotic and woody species (0–100% and 0–63%, respectively, Fig. 5). The 20 exotic plant species occur along a spectrum of invasiveness, although 90% are considered to have significant negative impacts in New Zealand conservation (75% of the 20 plant species)[80] or agricultural land (50%)[81]. Plants were grown from seed or cuttings collected from New Zealand's South Island (see Waller et al. 2020[55] for propagation details) and seedlings were randomly positioned in a ring, equally spaced around the centre of the pot during March 2017. Consistent positioning of plant species was used for replicates within each plant community, with plant communities replicated eight times to allow the application of herbivore and soil treatments (described below), and with replicates arranged together to minimise any environmental gradients.

To answer our research questions, we manipulated invertebrate herbivores (+Herbivore vs. −Herbivore) across mesocosm communities (Fig. 5). All mesocosms were covered with large mesh cages (Supplementary Fig. 13) (0.58 mm Cropsafe Mesh, 15% shade factor, Cosio Industries, Auckland, New Zealand) to keep added herbivores enclosed and deter most naturally occurring external herbivores (see Supplementary Methods for detailed description of cages). Herbivore populations were deliberately established in 80 mesocosms. Thirteen herbivore species that were added successfully established, along with seven self-colonising species, totalling 20 different species (establishment success and other herbivore species characteristics are detailed in Supplementary Table 25). These species were all polyphagous or oligophagous (see host ranges in Supplementary Table 25 and description of herbivore introductions in Supplementary Methods) and included seven native and 13 exotic herbivores from multiple feeding guilds (leaf and root chewers, suckers and miners). Each herbivore species was added to all +Herbivore mesocosms in equal density, regardless of whether a known host plant was present. Herbivore additions were staggered depending upon availability and some species were added multiple times to increase probability of establishment success and maintain populations (see Supplementary Methods for detailed description of protocols for each herbivore species). All self-colonising species were regularly removed from −Herbivore mesocosms, including spillover from intentional additions, but were allowed to establish populations in +Herbivore mesocosms. Several of the herbivore species produced multiple generations in the mesocosm communities (i.e., multiple life stages observed, or more individuals observed than were introduced), such as leafrollers, aphids, leafhoppers and slugs, and these are noted in Supplementary Table 25. Overall, our goal was not to replicate natural plant–herbivore communities, but to capture how native and exotic plants interact with a consistent suite of herbivores in novel

communities, the preference and performance of the herbivores and potential consequences for indirect effects. We complied with all relevant ethical regulations for animal testing and research; no formal ethics approval was required as invertebrate insect herbivores are not covered by ethics oversight in New Zealand.

The herbivore exclusion treatment was highly effective, reducing herbivore species presence on plants by 79% (generalised linear mixed model: $F_{1,585} = 584.68$, $P < 2.2e^{-16}$; Supplementary Fig. 14A), herbivore species biomass per plant by 84% (linear mixed model: $F_{1,137} = 651.55$, $P < 2.2e^{-16}$; Supplementary Fig. 14B), herbivore species richness per mesocosm by 59% (linear mixed model: $F = 152.10$, $P < 2.2e^{-16}$; Supplementary Fig. 14C), herbivore chewing and scraping damage per plant by 24% (generalised linear mixed model: $F = 276.22$, $P < 2.2e^{-16}$; Supplementary Fig. 14D), and PAC exerted and recieved by 98% (linear mixed model: $F_{1,139} = 342.64$, $P < 2.2e^{-16}$; Supplementary Fig. 14E) and 99.5% (linear mixed model: $F_{1,139} = 275.50$, $P < 2.2e^{-16}$; Supplementary Fig. 14F), respectively. Therefore, only data from the +Herbivore mesocosms were used to test our predictions, except for those relating to normalised degree and net herbivore impacts on plant biomass production (predictions 1b, 2a and 2b in Table 1; see statistical analyses below).

The herbivore treatment was crossed with a soil biota manipulation ('home' vs. 'away'), as part of another study[55] (Fig. 5). Soil biota was manipulated using a modified plant–soil feedback approach[48], where we grew each plant species in monoculture in 10 L pots of field-collected soil and pasteurised sand (50:50 mix) prior to the experiment to culture their associated soil biota. These conditioned soils were harvested after 9–10 months and used to create 'home' and 'away' soil inoculum mixtures for each plant community that were added to the mesocosms. 'Home' soils contained conditioned soils mixed from the eight species occurring in that community, and represent soils from an established invasion that contain both specialist and generalist soil biota. On the other hand, 'away' soils contained conditioned soils mixed from eight species occurring in one of the other 19 communities, but where a focal species did not occur. These 'away' soils represent previously uninvaded and thus contain no specialist soil biota. Therefore, although the soil treatment was not the main focus of this paper, it allowed us to test how specialist soil biota moderate plant–herbivore interactions in established versus new invasions, and we retained it as an explanatory variable in analyses to control for its potential effects.

**Data collection**. We measured herbivore richness, biomass, leaf damage by chewing and scraping herbivores and plant biomass (full list of response variables in Supplementary Table 26) (Fig. 5). Herbivores were surveyed on eight occasions: May, June, July, August, September and November in 2017 and January and April in 2018. For each survey, we counted the number of individuals of each herbivore species that were observed feeding on each plant. For species that reached high densities (e.g., aphids), abundance was estimated by surveying a portion of the plant and extrapolating to the entire plant. For some highly mobile or belowground herbivores it was difficult to reliably characterise feeding interactions through direct observation. For these species, we used restriction fragment length polymorphism (RFLP) to identify host plants with DNA extracted from frass, regurgitate or gut contents (see Supplementary Methods for detailed description of molecular protocols). Finally, because we could not practically measure the biomass of each individual herbivore from each mesocosm, we converted raw abundances to a standardised estimate of herbivore biomass for each species using mean dry biomass of a random sample of ten individuals. To calculate the mean biomass of each herbivore species for each individual plant, we multiplied the total abundance of the herbivore by its mean dry biomass per individual, and then divided by the number of times that plant was surveyed (plants that died were surveyed less than eight times). To estimate total mesocosm herbivore biomass, we multiplied the mean dry biomass per individual for each herbivore species with its total abundance across all surveys, and then summed across all herbivore species.

For each survey, we also assessed leaf damage by chewing and scraping herbivores on each plant against six different categories (0 = no damage, 1 = 1–5% leaf area chewed or scraped, 2 = 6–25%, 3 = 26–50%, 4 = 51–75%, 5 = >75%). We used these categories because of the large number of plants to survey and the difficulties of non-destructively measuring percent leaf area removal at finer resolution in situ. We obtained an overall estimate of damage throughout the experiment by transforming the categories to median percent damage values (e.g., category 3 = 38%) and calculating mean percent damage per survey for each plant. Finally, plants were harvested after 1 year, above- and belowground biomass separated and washed, dried at 65 °C, and weighed. Additional methodological details are described in Supplementary Methods and Waller et al. (2020)[55].

**Data analysis**. For each response variable, we used (generalised) linear mixed effects models to ask whether native and exotic plants (and native-dominated and exotic-dominated communities) differed in their direct (predictions 1a–c and 4a in Table 1), indirect (predictions 3a and 4b) or net (predictions 2a, b) interactions with herbivores and soil biota. For analyses at the individual plant level, each model included plant provenance (native, exotic), the soil treatment ('home', 'away'), and their interaction as fixed effects (Supplementary Table 26 contains model structure details), with plant species and mesocosm nested within plant community as random effects. To assess how herbivores influenced the biomass production of native and exotic plants, we used data from all mesocosms and included the

herbivore treatment and its interactions in the model. Post hoc pairwise contrasts involving more than two treatment combination levels (i.e., interactions) were conducted using Bonferroni corrected Tukey tests. For analyses at the mesocosm level, each model included the proportion of exotic species planted in the community (0–100%), the soil treatment and their interaction as fixed effects, with plant community as a random effect (mesocosm was nested within plant community for analyses of herbivore biomass and herbivore:plant biomass ratio to account for the non-independence of native and exotic herbivores occurring on the same plant). For analyses of herbivore species' presence, herbivore biomass, and herbivore:plant biomass ratio, the herbivore provenance (native, exotic) was also included as a fixed effect and herbivore species and mesocosm nested within plant community as random effects.

The number of observations and model error distributions used varied depending upon the response variable and some response variables were transformed to meet model assumptions (summarised in Supplementary Table 26). For analyses of herbivore presence, we retained absent interactions (i.e., zeroes in the data) that were within the fundamental host range for each herbivore species (based on the experiment-wide meta-web; i.e., the herbivore species fed on the focal host in at least one mesocosm) and discarded data for those that were not. Herbivore biomass was assessed using a two-stage model, where we first examined treatments that were influential to the presence or absence of herbivore species on plants within their fundamental host range, followed by secondary analyses to assess herbivore biomass only on plants where herbivores were present. Herbivore presence was modelled using a binomial error distribution, while herbivore biomass was log-transformed and modelled using a normal error distribution. Normalised degree did not require transformation and was modelled using a normal error distribution. Herbivore species richness per mesocosm was also untransformed and was modelled using a Poisson error distribution. Percent leaf damage from chewing and scraping invertebrate herbivores was analysed using a gamma error distribution with a log link function, and was logit-transformed before a constant of 5 was added to conform to the gamma distribution. Both measures of PAC below were log-transformed and modelled using a normal error distribution. Dead plants were excluded from analyses of plant biomass, which was log-transformed and modelled using a normal error distribution.

For all plausible models, Cook's D and quantile-quantile plots were used to identify potentially influential data points. However, in no case did removal of these data points qualitatively change model conclusions, thus we retained them in analyses. All model assumptions were tested for and satisfied, and Poisson and binomial models were checked for overdispersion, with none detected. We report estimated marginal means and standard errors from fitted models, back-transformed when appropriate.

We used normalised degree (i.e., the proportion of herbivore species that fed upon a given host plant out of the total herbivore species in the mesocosm) to quantify herbivore richness for each plant, because the number of invertebrate species that established varied among mesocosms. Measuring the plant–herbivore interactions of the entire community allowed us to estimate each species' potential for apparent competition (PAC). PAC is a metric devised by Müller et al. (1999)[86] that describes the sharing of interaction partners between two species in a community, and has been previously used to predict outcomes of indirect interactions in host–parasitoid communities[43–45]. To estimate PAC for each host plant species pair in a given mesocosm, we calculated $d_{ij}$, the proportion of herbivore biomass attacking plant species $i$ that is shared with plant species $j$. In the equation for pairwise PAC below (see also Fig. 5), $\alpha$ represents link strength (i.e., herbivore biomass), $i$ and $j$ are the focal pair of host plant species, $m$ is all plant species from 1 to $H$ (the number of plant species in the community), $k$ is a herbivore species, and $l$ is all herbivore species from 1 to $P$ (the number of herbivore species in the community)[86].

$$d_{ij} = \sum_{k=1}^{P}\left[ \frac{\alpha_{ik}}{\sum_{l=1}^{P}\alpha_{il}} \frac{\alpha_{jk}}{\sum_{m=1}^{H}\alpha_{mk}} \right] \quad (1)$$

After calculating pairwise PAC between all plants within each mesocosm, we quantified the potential for focal species $i$ to exert apparent competitive effects (PAC_exerted) by summing PAC values for the focal species on all other community members (excluding intraspecific PAC; PAC = 0 if plants shared no herbivores). We also quantified the potential for focal species i to receive apparent competitive effects (PAC_received) by summing pairwise PAC values from all other community members to the focal plant. Because PAC should vary with the total number of herbivores in the community, but was calculated on a standardised scale within each mesocosm (i.e., using the relative strength of interactions), we weighted community-level PAC values using the total herbivore biomass of the focal plant (for PAC_exerted) or the rest of the community (for PAC_received). We used these data to examine potential causes and consequences of PAC, asking whether: (1) exotic plants had greater PAC_exerted and lower PAC_received than native plants (prediction 3a in Table 1); (2) plants with greater PAC_received had lower total biomass and higher herbivore damage (prediction 3b); and (3) larger plants had greater PAC_exerted (prediction 3c). Hypotheses were tested using linear mixed models. Response variables were transformed as per Supplementary Table 26 and plant species and mesocosm nested within plant community were included in the models as random effects.

Finally, to explore whether plant–herbivore interactions contributed to the exotic plant dominance of plant communities (prediction 2b in Table 1), we asked whether the proportion of realised exotic biomass differed from the expected value based on the proportion of exotic plant species planted in the community. We calculated the proportion of exotic plant biomass per mesocosm and estimated the mean and 95% confidence interval for each level of proportion of exotic species planted in the community (i.e., 25, 50 and 75% exotic, but excluding communities planted with 0 and 100% exotic species) crossed with each level of the herbivore treatment. We then assessed whether 95% confidence intervals overlapped levels of the proportion of exotic plant species planted in the community (i.e., greater dominance by exotic plants than expected) and if 95% confidence intervals overlapped for +Herbivore vs. −Herbivore mesocosms within each level of proportion of exotics planted (i.e., herbivores altered the dominance of exotic plants). All analyses were performed in R 3.6.1[87] using the lme4[88], emmeans[89] and bipartite[90] packages.

**Reporting Summary**. Further information on research design is available in the Nature Research Reporting Summary linked to this article.

## Data availability
Data available for download from Dryad: https://doi.org/10.5061/dryad.0vt4b8gzd.

## Code availability
Code is available on request from the corresponding author.

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

## Acknowledgements

We are grateful to Jim Allen, Neroli Allen, Anne Barrington, Jennifer Bufford, David Conder, Leo Condron, Daniel Dash, Lochlan Dickie, Colin Ferguson, Filipe França, Travis Glare, Joanna Green, Andrew Holyoake, John Hunt, David Jack, Nina Koele, Brian Kwan, Stuart Larsen, Zach Marion, Francesco Martoni, Aimee McKinnon, Leona Meachen, Tara Murray, Kate Orwin, Alexandra Puértolas, John Ramana, Brent Richards, Sarah Richardson, Ralph Scott, Marcus-Rongowhitiao Shadbolt, Georgia Steel, Jono Tonkin, Ralph Wainer, Dean Waller, Mark Waller, Angela Wakelin, Steve Wakelin and Sandy Wilson for support, technical and otherwise. Thanks also to the ABiNZE, Ecosystem Mycology, Stouffer, Tylianakis and Weed Wing research groups for helpful discussions about the experiment, and to Angela Moles and Tina Heger for helpful comments on an earlier version of the manuscript. This project was supported by Centre of Research Excellence funding to the Bio-Protection Research Centre from the Tertiary Education Commission of New Zealand.

## Author contributions

I.A.D. and J.M.T. obtained funding. All authors designed the experiment, W.J.A. and B.I. P.B. implemented the herbivore treatment and W.J.A. and L.P.W. conducted the experiment and collected data. W.J.A. led analyses and wrote the first draft of the manuscript, and all authors contributed substantially to revisions.

## Competing interests

The authors declare no competing interests.
