## [Peer Review File · Nature Communications]

Reviewers' Comments:

Reviewer #1:

Remarks to the Author:

The current manuscript is a very well written and exhaustively performed comparison of herbivory (specialist, generalist and exotic) on 19 native and 20 exotic plant species in New Zealand. In contrast with expectations, the authors found larger herbivore loads on exotic plants than on natives, with heavier herbivore biomass and marginally greater levels of plant tissue damage.

The strong point of the paper is that it adds an important element to our understanding of the various hypotheses that underpin alien invasive plant success in novel ecosystems. On this point alone the paper will be well-cited and act as a framework for future studies of this kind. On the other hand, I have some serious points that need to be addressed in a revised version; I know the authors are word-limited but I hope they can accommodate these points.

First of all, it is well-known that 98% (or thereabouts) of exotic species do not become ecologically disruptive invaders in their new ranges. By 'disruptive' of course I mean that they do not displace native plants or plant assemblages or have multitrophic-level effects that are well described in the literature (this is described in papers by Jim Cronin with i.e. spartina and brome grass and in the Harvey et al. [2014] review in Annual Review of Entomology that the authors did not cite but honestly should as it contains many examples that run counter to their findings).

The question that needs to be addressed here is how many of the 20 exotic plant species actually fall into the category of being 'ecologically disruptive'? In other words, how many are like plants species such as garlic mustard, Russian olive, yellow-star-thistle, cheat-grass, kudzu vine etc. that are hugely disruptive alien invaders in the United States? These plants clearly possess traits that render them more likely to become highly invasive. Garlic mustard, *Alliaria petiolata*, for instance, has a novel allelochemistry which is toxic to many native north American insect herbivores and is also allelopathic to soil mutualists of native plant species. My personal experience is that herbivores in North America virtually ignore it completely. Another paper the authors have not cited but should (Cappuccino and Arnason, 2006) examines the importance of novel chemical defences as evidence of reasons for plants becoming invasive pests. Contrast this with work of Louda and colleagues, who found that the fast-growing invasive thistle *Cirsium arvense* is heavily attacked by native North American herbivores that simply switch from the native bull thistle, *C. altissimum*, to *C. arvense*. They clearly play a role in suppressing the invader. The authors therefore need to explain how many of the plant species they are studying here are more like a garlic mustard than a *C. arvense* - in other words, how many are truly novel and have become or are poised to become highly invasive in New Zealand?

The second point I wish to make is for the authors to explain how they delineated plants as being 'generalists' or 'specialists'. Recent work by Hugh Loxdale and colleagues suggests that the vast majority of insect herbivores are, to a greater or lesser degree, specialists (they posit this as 80% or more). The co-evolutionary arms race between specialists and their food plants is leaving the generalists behind, and even so-called generalists (e.g. the Noctuidae) are 'composite-specialists', where genotypes tend to prefer certain plants and avoid others. Thus, at the species level they may feed on multiple plant species, but certain genotypes evolve to prefer certain food plants with phylogenetically-conserved traits, a pre-requisite for speciation and specialisation. Can the authors therefore so easily separate real specialists and generalists?

The third point I wish to make is that the authors found very little damage to the exotic and native plants (4.3% mean). Given that plants often exhibit the ability to recover from much more extensive damage (e.g. resilience), how much do the authors think that this plays a role here in determining the success of invaders in novel New Zealand ecosystems?

The experiments were carried out in mesocosms. How much might natural heterogeneity alter the

ability of insects to locate the plants and thus alter patterns of herbivory? Neighbourhood effects are important in terms of physical and chemical barriers or impediments to plant-finding ability in insect herbivores.

The points I make here are guidance to make the paper more comprehensive because the authors forcefully conclude that enemy-release (with respect to insects) is not an important factor in invasive success; I believe that there are many caveats that need to be at least considered before really making firm conclusions when testing various hypotheses describing the success or failure of invasives. I do like the fact that the authors highlight the importance of soil pathogens - by now we know that they are a potentially very important factor.

None of my comments are meant to take away from the importance of the paper - it is certainly publishable if the referees incorporate the points I mention above.

Reviewer #2:

Remarks to the Author:

Review for nature communications, Manuscript NCOMMS-20-41158

Exotic plants accumulate and share generalist herbivores but dominate communities via rapid growth

General comments

This manuscript describes results of a large mesocosm experiment with a very complex design. The results are interpreted with reference to several hypotheses about interactions between native and exotic plants and herbivores. This combination of a carefully designed experiment including an exceptionally large set of species, with a careful and complex discussion based on theory is impressive. I appreciate Table S1.1., corroborating that this study really is exceptional, because it addresses the question whether there is a release of plants from herbivorous enemies, and at the same time looks at whether / how a potential release affects the performance of the exotic plants in an experimental community. I have several suggestions for improvement, mainly connected to the presentation of results.

My main concern is that so far, the high complexity of the study leads to confusion. The research questions, conceptual figure (Fig. 1A) and the discussion section should be tied together more closely, and be more efficiently used to better explain the complex analyses and results.

In principle, I like the idea to split the research questions in detailed predictions relating to the expected results, as done in Table 1. However, in the current version this does not really work out: The relation of the sub-questions to the main question is not always clear, and several aspects have not been introduced and motivated in the text (see detailed comments below). Also, so far the results and discussion section are not well connected to these questions.

Also, Fig. 1 A is nice and explains the concepts well, but so far is not very well connected to the research questions (there, neither the enemy of my enemy nor the invasional meltdown hypothesis are mentioned), nor are all of these hypotheses mentioned again in the discussion section. I suggest changing the figure to better represent the research questions (even if then, there is no hypothesis to match that question).

Detailed comments

With the method section moved to the back (according to nature communication format), the numbering of Figures and Tables (in the supplement) should be changed accordingly. Fig. 1A should become Fig. 1, and the rest of this figure should be moved to the methods section.

Title: maybe add 'but still dominate..' to emphasize this interesting finding a little more

Line 23: I suggest 'hamper' instead of 'resist'

Line 29: This is unclear, maybe change to 'indirect interactions with herbivores'? Or 'differ in their interactions with herbivores'? Or do you mean indirect interactions with each other?

Lines 103-104: 'in the introduced range' can be removed here, I suggest (it gives no additional information but instead may confuse readers)

Line 111: Why should exotic plant species interact stronger with exotic herbivores than native plants? This is not in line with the idea of biotic resistance, and I suggest deleting the bracket (also in Table 1). Also, in the introduction (lines 77 to 101) many potential effects of possible interactions are discussed, but not this one. What is the basis for this prediction? The only thing I could think of would be that the exotic herbivore is familiar with the exotic plant as food resource because these species co-occur in their native ranges, but that for some reason the exotic plant did not evolve any defenses. I don't think this hypothesis has a name yet. The inverse of this hypothesis (natives are less affected by exotic herbivores than exotics) has no name either, but could e.g. be called the 'inverse enemy release hypothesis'. It is not introduced in the main text yet either.

Table 1: Question 1 d) has not been introduced in the main text, and it is unclear how it relates to question 1. You could consider removing it from the hypotheses, and state somewhere that the reproductive status may somehow confound the results (give a reason why) and you therefore check for potential effects

Lines 113-115 and Table 1: It is unclear what is meant by 'proportional impact'(line 114), and what is the difference to 'damage' (line 110). Correspondingly, I do not understand the difference between Question 1c) and 2a). After having read the results section, I suggest talking in general about 'reduced biomass production' instead of 'proportional impact'.

Table 1 Question 2b): Something is wrong with this sentence – please re-formulate (I don't get its meaning)

Table 1 question 3: It is not clear how questions b and c relate to the overall question. Also, these two hypotheses have not been introduced in the main text, and it is therefore not clear what is their theoretical basis. Maybe you could similarly to above state that biomass could have a confounding effect and you check it therefore (i.e. not as a specific research hypothesis)

Line 122: It has not been mentioned before that the soils have been treated. Please add this information to the introduction.

Line 128-130: This sentence is unclear. Do you mean: "High total herbivore biomass could amount to proportionally low herbivore biomass on exotics, if exotic plants had higher biomass"?

Lines 137-139: In the data collection section, it is not described that you assessed which species were reproducing. How did you do that?

Line 156: Was damage on average almost double?

Fig. 2 G and H should be removed from Fig. 2, and turned into a Fig. 5, since they are treated only after Fig. 3 and 4

Line 164-165: See comments above: if here, it is about biomass produced and not biomass removed by herbivores, this should be made much clearer. I suggest not to refer to this as 'impact on biomass', but maybe rather 'reduction in biomass production in mesocosms with herbivores'

Line 166: I suggest reformulating to "Exotic plants produced/had 31% less biomass in ...". Otherwise this effect can be confused with the effects of herbivores removing biomass by chewing.

Fig. 3B: I suggest removing this fig. from the main text, since it is not related to any of the research questions.

Line 208: This sounds like you checked for the growth -rates – please reformulate (or present respective results)

Lines 238: Please discuss somewhere what this reduction in biomass may mean: I guess it is not only biomass lost by chewing, but also reduced growth? What about belowground effects? See also my comments above

Line 493: In the data collection section, it is not described that you assessed which species were reproducing. How did you do that?

Reviewer: Tina Heger

ORCID <https://orcid.org/0000-0002-5522-5632>

Reviewer #3:

Remarks to the Author:

First, I would like to apologize for reviewing this again – the change of title, combined with the fact that this is my busiest teaching time led me to accept the review quickly without realizing I had reviewed it before – I would have stepped back had I realized (not really fair for you to come up against the same reviewer more than once!). Having said that, I think the revisions you have made are fantastic. This is now a beautifully written paper that presents the results from an extremely impressive experiment on a very important topic. I think it will be of high interest to many ecologists and land managers worldwide.

You have done an amazing job of responding to all my original comments. I have only two remaining suggestions:

1. You have discussed the fact that differences between native and introduced species in traits like N-fixing ability and SLA might affect your findings (thank you!). However, the reader is still left wondering whether the higher rates of herbivory on exotic species might be attributable to these factors, or if the species' provenance is actually important in its own right (which is an important question for accurate interpretation of your findings). I don't think this would be a difficult analysis (one might compare herbivory ~ traits with herbivory ~ traits * provenance and ask whether the model that included provenance explained significantly more of the variation than did the model with traits alone).

2. I wondered why you didn't make more of the soil treatment in your manuscript – it is pretty impressive and unusual for a study to simultaneously consider above-ground and below-ground interactions. It may be a null result, but it is still interesting and important.

Best wishes,
Angela Moles

Reviewer 1

The current manuscript is a very well written and exhaustively performed comparison of herbivory (specialist, generalist and exotic) on 19 native and 20 exotic plant species in New Zealand. In contrast with expectations, the authors found larger herbivore loads on exotic plants than on natives, with heavier herbivore biomass and marginally greater levels of plant tissue damage. The strong point of the paper is that it adds an important element to our understanding of the various hypotheses that underpin alien invasive plant success in novel ecosystems. On this point alone the paper will be well-cited and act as a framework for future studies of this kind. On the other hand, I have some serious points that need to be addressed in a revised version; I know the authors are word-limited but I hope they can accommodate these points.

Thank you for your positive words and constructive comments about our manuscript.

First of all, it is well-known that 98% (or thereabouts) of exotic species do not become ecologically disruptive invaders in their new ranges. By 'disruptive' of course I mean that they do not displace native plants or plant assemblages or have multitrophic-level effects that are well described in the literature (this is described in papers by Jim Cronin with i.e. spartina and brome grass and in the Harvey et al. [2014] review in Annual Review of Entomology that the authors did not cite but honestly should as it contains many examples that run counter to their findings). The question that needs to be addressed here is how many of the 20 exotic plant species actually fall into the category of being 'ecologically disruptive'? In other words, how many are like plants species such as garlic mustard, Russian olive, yellow-star-thistle, cheat-grass, kudzu vine etc. that are hugely disruptive alien invaders in the United States? These plants clearly possess traits that render them more likely to become highly invasive. Garlic mustard, *Alliaria petiolata*, for instance, has a novel alleochemistry which is toxic to many native north American insect herbivores and is also allelopathic to soil mutualists of native plant species. My personal experience is that herbivores in North America virtually ignore it completely. Another paper the authors have not cited but should (Cappuccino and Arnason, 2006) examines the importance of novel chemical defences as evidence of reasons for plants becoming invasive pests. Contrast this with work of Louda and colleagues, who found that the fast-growing invasive thistle *Cirsium arvense* is heavily attacked by native North American herbivores that simply switch from the native bull thistle, *C. altissimum*, to *C. arvense*. They clearly play a role in suppressing the invader. The authors therefore need to explain how many of the plant species they are studying here are more like a garlic mustard

than a *C. arvense* - in other words, how many are truly novel and have become or are poised to become highly invasive in New Zealand?

Thank you for this thoughtful comment. Of the exotic plants included in our study, 90% are considered to be invasive weeds (i.e., ‘ecologically disruptive’) in either conservation or agricultural land in New Zealand (based on lists published by Howell 2008 and Ghanizadeh & Harrington 2019). We have now added the ‘weed status’ of each exotic plant to Supplemental Table S4.2. Furthermore, we also now describe the proportion of exotic plants that are considered to be invasive weeds as part of the Methods section (“The 20 exotic plant species occur along a spectrum of invasiveness, although 90% are considered to have significant negative impacts in New Zealand conservation (75% of the 20 plant species)⁸⁰ or agricultural land (50%)⁸¹.”; L422-425), and also discuss our results in this context (L346-353).

We have also incorporated references suggested by the reviewer (Bezemer et al. 2014; Cappuccino & Arnason 2006; Eckberg et al. 2012) as well as others (Liu et al. 2007; Keeler & Chew 2008; Heinen et al. 2018) into the text, and apologise for these oversights. Although we now have more than the recommended number of 70 references, we believe that their inclusion is justified by the broad research topics that motivate our manuscript and can be offset by the shorter length of other parts of the paper, such as the 6 display items (maximum of 10) and 6871 total words (maximum of 8000 words for main text and methods combined).

Bezemer, T.M., Harvey, J.A. & Cronin, J.T. Response of native insect communities to invasive plants. *Ann. Rev. Entomol.* **59**, 119–141 (2014).

Cappuccino, N. & Arnason, J.T. Novel chemistry of invasive exotic plants. *Biol. Lett.* **2**, 189–193 (2006).

Eckberg, J.O., Tenhumberg, B. & Louda, S.M. Insect herbivory and propagule pressure influence *Cirsium vulgare* invasiveness across the landscape. *Ecology* **93**, 1787–1794 (2012).

Ghanizadeh, H. & Harrington, K.C. Weed management in New Zealand pastures. *Agronomy* **9**, 448 (2019).

Heinen, R., Biere, A., Harvey, J.A. & Bezemer, T.M. Effects of soil organisms on aboveground plant-insect interactions in the field: patterns, mechanisms and the role of methodology. *Front. Ecol. Evol.* **6**, 106 (2018).

Howell, C. *Consolidated list of environmental weeds in New Zealand*. DOC Research & Development Series 292 (Department of Conservation, Wellington, New Zealand, 2008).

Keeler, M.S. & Chew, F.S. Escaping an evolutionary trap: preference and performance of a native insect on an exotic invasive host. *Oecologia* **156**, 559–568 (2008).

Liu, H., Stiling, P. & Pemberton, R.W. Does enemy release matter for invasive plants? evidence from a comparison of insect herbivore damage among invasive, non-invasive and native congeners. *Biol. Invasions* **9**, 773-781 (2007).

The second point I wish to make is for the authors to explain how they delineated plants as being 'generalists' or 'specialists'. Recent work by Hugh Loxdale and colleagues suggests that the vast majority of insect herbivores are, to a greater or lesser degree, specialists (they posit this as 80% or more). The co-evolutionary arms race between specialists and their food plants is leaving the generalists behind, and even so-called generalists (e.g. the Noctuidae) are 'composite-specialists', where genotypes tend to prefer certain plants and avoid others. Thus, at the species level they may feed on multiple plant species, but certain genotypes evolve to prefer certain food plants with phylogenetically-conserved traits, a pre-requisite for speciation and specialisation. Can the authors therefore so easily separate real specialists and generalists?

In addressing this comment, we are assuming that the reviewer meant “herbivores” instead of “plants” in the first sentence. We agree with the reviewer that describing the herbivores used in our experiment as “specialist” or “generalist” was lacking in nuance. Therefore, we have replaced these terms throughout the manuscript with “monophagous”, “oligophagous”, and “polyphagous” (all defined in the main text; L48-52), as suggested by Loxdale et al. (2019). However, we retain the use of specialist and generalist when referring to soil biota and pathogens, as terms ending in “-phagy” (implying feeding) cannot be applied to plant-soil biota interactions.

We have also added a column to Table S4.3 that quantifies the number of host plant species that each herbivore was observed feeding on in the mesocosm experiment. These data show that all of the herbivores that established in >10 mesocosms also fed on at least seven different host plant species, with the exception of the *Sitona* weevils, which are well-known oligophages. Because we could not reliably infer the host range of herbivore species that established in only a few mesocosms, we also used the Plant-SyNZ Database (the most comprehensive database of plant-insect interactions observed in New Zealand: <https://plant-synz.landcareresearch.co.nz>) to add details to Appendix 3 about the known host range of each herbivore species in New Zealand. Based on these published and observed host ranges, we feel confident in attributing the vast majority of herbivory observed in the mesocosms to polyphagous herbivore species, and thus we interpret the results in this context.

Although we agree with the reviewer that herbivore individuals or genotypes within a species may differ in their preferred host plants, we did not set out to address this question with our study. Rather, we attempted to minimize this genotypic variation by introducing herbivores to mesocosms that were sampled from a single population for most of the herbivore species.

Loxdale, H.D., Balog, A. & Harvey, J.A. Generalism in nature... The great misnomer: aphids and wasp parasitoids as examples. *Insects* **10**, 314 (2019).

The third point I wish to make is that the authors found very little damage to the exotic and native plants (4.3% mean). Given that plants often exhibit the ability to recover from much more extensive damage (e.g. resilience), how much do the authors think that this plays a role here in determining the success of invaders in novel New Zealand ecosystems?

We agree that 4.3% of leaf tissue damaged is low, although this level of damage is still close to the observed average of around 7.5% across the plant Kingdom (Kozlov et al. 2015).

Despite this low level of chewing and scraping damage, we found that exotic plants produced 31% less biomass in mesocosms with added herbivores compared to those with reduced herbivores, which we consider to be a direct measurement of the 'net' herbivore impact on plants. This finding suggests that we may be underestimating herbivore damage, perhaps because this measure doesn't account for damage that cannot be observed (e.g., completely excised leaves or belowground herbivores), damage or disease transmission by non-chewing herbivores (e.g., aphids and other sucking insects), or reduced growth of impacted plants (e.g., when young leaves are damaged). We have now revised the discussion to make this

interesting comparison between percent herbivore damage and proportional reductions in biomass production: “Average damage to plant tissue from chewing and scraping herbivores was only 4.3% of leaf tissue removed and this did not differ between native and exotic plants. However, this level of damage is similar to the average of 7.5% observed across the plant Kingdom⁵⁶, and herbivore damage can translate to variable impacts on plant fitness, from complete defoliation and death through to tolerance and overcompensatory growth. ... Thus, we consider the 7-fold larger effect of the herbivore treatment on biomass production compared to chewing damage to be a more direct measure of the net impact of herbivores (i.e., the sum of biomass lost from chewing herbivores, unquantified damage from sucking insects and belowground herbivores, and reduced growth of impacted plants).” (L273-286).

We agree that herbivory can have variable impacts on plant fitness, from complete defoliation and death through to tolerance and overcompensatory growth. However, because exotic plants experienced reduced biomass production in mesocosms with added herbivores, whereas native plants did not, we found no evidence for higher tolerance of herbivory for exotic compared to native plants: “Finally, given that exotic but not native plants experienced reduced biomass production in mesocosms with added herbivores, we conclude that the native plants in our experiment were more tolerant of herbivory than exotic plants, supporting the findings of some studies^{57,58} but not others^{30,59}.” (L286-289).

Kozlov, M.V., Lanta, V., Zverev, V. & Zvereva, E.L. Global patterns in background losses of woody plant foliage to insects. *Glob. Ecol. Biogeogr.* **24**, 1126–1135 (2015).

The experiments were carried out in mesocosms. How much might natural heterogeneity alter the ability of insects to locate the plants and thus alter patterns of herbivory? Neighbourhood effects are important in terms of physical and chemical barriers or impediments to plant-finding ability in insect herbivores.

Our experiment incorporated some natural heterogeneity in that each plant species generally had multiple different neighbours across the 20 different mesocosm communities, and natural mortality within mesocosms would have only further increased the diversity of neighbours for each plant species. Thus, neighbourhood effects would have added to the error variance and contributed to the substantial variability in plant-herbivore interactions that was observed in the experiment. However, the reviewer’s point is fair and we address how our results may translate to natural systems (including the importance of neighbourhood effects) in the

discussion section: “We must also acknowledge that our mesocosm communities did not replicate natural communities, which are almost certainly affected by greater herbivore diversity, indirect effects of natural enemies (i.e., predators and parasitoids)^{66,67}, and herbivore aggregation, heterogeneity and neighborhood effects over larger spatial scales⁶⁸. Thus, our results should be taken with caution when translating to natural systems.” (L312-316). We have also added a photograph of some example mesocosms to aid the reader in visualising the plant communities (Figure S4.1).

The points I make here are guidance to make the paper more comprehensive because the authors forcefully conclude that enemy-release (with respect to insects) is not an important factor in invasive success; I believe that there are many caveats that need to be at least considered before really making firm conclusions when testing various hypotheses describing the success or failure of invasives. I do like the fact that the authors highlight the importance of soil pathogens - by now we know that they are a potentially very important factor. None of my comments are meant to take away from the importance of the paper - it is certainly publishable if the referees incorporate the points I mention above.

Thank you for these positive thoughts on our manuscript. We have taken your comments on board and have broadened the discussion of the results to address appropriate caveats (as outlined in detail above), and have also further expanded the discussion of the potential role of soil biota (see comments in response to other reviewers below).

Reviewer 2

General comments

This manuscript describes results of a large mesocosm experiment with a very complex design. The results are interpreted with reference to several hypotheses about interactions between native and exotic plants and herbivores. This combination of a carefully designed experiment including an exceptionally large set of species, with a careful and complex discussion based on theory is impressive. I appreciate Table S1.1., corroborating that this study really is exceptional, because it addresses the question whether there is a release of plants from herbivorous enemies, and at the same time looks at whether / how a potential release affects the performance of the exotic plants in an experimental community.

Thank you for these positive words about our experiment and manuscript.

I have several suggestions for improvement, mainly connected to the presentation of results.

My main concern is that so far, the high complexity of the study leads to confusion. The research questions, conceptual figure (Fig. 1A) and the discussion section should be tied together more closely, and be more efficiently used to better explain the complex analyses and results.

We have now improved the cohesiveness of the manuscript by clarifying the conceptual links between the background literature, research questions and predictions, results and tables/figures, and discussion. Specific examples of the changes made are listed in the specific comments below.

In principle, I like the idea to split the research questions in detailed predictions relating to the expected results, as done in Table 1. However, in the current version this does not really work out: The relation of the sub-questions to the main question is not always clear, and several aspects have not been introduced and motivated in the text (see detailed comments below). Also, so far the results and discussion section are not well connected to these questions.

We agree with the reviewer that some predictions in Table 1 were not adequately linked to the key research questions or the introduction. To address these concerns, we have: 1) removed some predictions from Table 1 (i.e., prediction 1d) in favour of including them in the main text (also see response to comment below regarding this prediction); 2) clarified the theory and motivation behind other hypotheses/predictions retained in the table (also see response to more detailed comments below); and 3) better linked our results back to theory in the discussion.

Also, Fig. 1 A is nice and explains the concepts well, but so far is not very well connected to the research questions (there, neither the enemy of my enemy nor the invasional meltdown hypothesis are mentioned), nor are all of these hypotheses mentioned again in the discussion section. I suggest changing the figure to better represent the research questions (even if then, there is no hypothesis to match that question).

We have now better linked Figure 1 with the research questions and the main text. Specifically, Figure 1 now only includes diagrams that illustrate our research questions/predictions that are outlined in the text at the end of the introduction and in Table 1. We also refer to these diagrams when we introduce the research questions (e.g., L138, 139, 142, 143, 145, 147). Furthermore, we revisit how our results support or refute the invasion

hypotheses that motivated our predictions throughout the discussion (e.g., L247, 272, 301, 354, etc.).

Detailed comments

With the method section moved to the back (according to nature communication format), the numbering of Figures and Tables (in the supplement) should be changed accordingly.

Thank you for noticing this error. We have now fixed the numbering of figures and tables throughout the main text and supplement.

Fig. 1A should become Fig. 1, and the rest of this figure should be moved to the methods section.

We have now separated this figure into two separate figures – one for the hypotheses (Fig. 1, introduction) and one for the experimental design and analyses (Fig. 6, methods).

Title: maybe add ‘but still dominate.’ to emphasize this interesting finding a little more
Changed as the reviewer suggests.

Line 23: I suggest ‘hamper’ instead of ‘resist’
We have changed to “impede”.

Line 29: This is unclear, maybe change to ‘indirect interactions with herbivores’? Or ‘differ in their interactions with herbivores’? Or do you mean indirect interactions with each other?
Changed to “differ in their interactions with herbivores” as suggested.

Lines 103-104: ‘in the introduced range’ can be removed here, I suggest (it gives no additional information but instead may confuse readers)
Changed as suggested.

Line 111: Why should exotic plant species interact stronger with exotic herbivores than native plants? This is not in line with the idea of biotic resistance, and I suggest deleting the bracket (also in Table 1). Also, in the introduction (lines 77 to 101) many potential effects of possible interactions are discussed, but not this one. What is the basis for this prediction? The only thing I could think of would be that the exotic herbivore is familiar with the exotic plant as food resource because these species co-occur in their native ranges, but that for some

reason the exotic plant did not evolve any defenses. I don't think this hypothesis has a name yet. The inverse of this hypothesis (natives are less affected by exotic herbivores than exotics) has no name either, but could e.g. be called the 'inverse enemy release hypothesis'. It is not introduced in the main text yet either.

When asking how exotic herbivores differ in their interactions with native and exotic plants, multiple alternate predictions can emerge from invasion theory: 1) native plants have stronger interactions, which may be expected if exotic plants experience 'enemy release' because they possess co-evolved defences that are lacking for 'naïve' native plants (this is a version of the 'enemy of my enemy' hypothesis, which we had previously outlined in the third paragraph of the introduction and in Figure 1); 2) exotic plants have stronger interactions, which may be expected if native plants possess 'novel weapons' that deter attack by exotic herbivores; or 3) no difference in interactions with herbivores between native and exotic plants. We have now added these competing hypotheses to the third paragraph of the introduction ("For example, the enemy of my enemy hypothesis posits that co-introduced enemies should cause greater harm to native than exotic species, based on the potential lack of co-evolved defenses^{5,32}. Alternatively, exotic herbivores could cause greater harm to exotic than native species if native plants possess defences that are novel to exotic herbivores¹¹."; L84-88) and no longer refer to exotic herbivores preferring to feed on exotic plants as 'biotic resistance' in the manuscript, as requested by the reviewer.

Table 1: Question 1 d) has not been introduced in the main text, and it is unclear how it relates to question 1. You could consider removing it from the hypotheses, and state somewhere that the reproductive status may somehow confound the results (give a reason why) and you therefore check for potential effects

Following the reviewer's recommendation, we have now removed this prediction/question from Table 1 and instead state the rationale and findings of trait-based analyses in the discussion section: "To further understand how traits may have mediated differences in plant-herbivore interactions (i.e., herbivore presence, biomass, diversity, and damage to plants) between native and exotic plant species, we quantified whether variation in these response variables could be explained by the main effects and interactions of plant provenance with several traits of plants and herbivores (see Appendix S3 for details on these analyses). However, we found no consistent relationships between traits and plant-herbivore interactions, with results depending upon the response variable and trait being investigated (see Appendix S3 for detailed results)." (L339-346) (see also response to first comment by

Reviewer 3). This change now means that the specific predictions in Table 1 are clearly linked to the corresponding overarching research question.

Lines 113-115 and Table 1: It is unclear what is meant by 'proportional impact' (line 114), and what is the difference to 'damage' (line 110). Correspondingly, I do not understand the difference between Question 1c) and 2a). After having read the results section, I suggest talking in general about 'reduced biomass production' instead of 'proportional impact'.

We agree with the reviewer and have now been more specific throughout the manuscript. We now avoid using the vague term "impacts" when describing our results and have instead been more descriptive, using "proportional reductions in biomass production from herbivores".

Table 1 Question 2b): Something is wrong with this sentence – please re-formulate (I don't get its meaning)

We have rewritten this prediction for clarity: "Exotic plants make up a disproportionate proportion of plant community biomass, especially when herbivores are present".

Table 1 question 3: It is not clear how questions b and c relate to the overall question. Also, these two hypotheses have not been introduced in the main text, and it is therefore not clear what is their theoretical basis. Maybe you could similarly to above state that biomass could have a confounding effect and you check it therefore (i.e. not as a specific research hypothesis)

Prediction 3c (now prediction 3b) is directly related to the major research question, where we ask whether plants with high potential to receive strong potential for apparent competition had lower biomass at the end of the experiment. However, we agree with the reviewer that this was not clear in the previous version of the manuscript and so we have now expanded the research question and been more explicit about this analysis in the methods (L576-580) and results sections (L225-233). Although we acknowledge that prediction 3b (now prediction 3c) is not directly related to the overarching research question, we have retained this prediction in the table because it allows for Table 1 (showing research questions and predictions) to be fully linked with Table S4.4 (showing model structure), and for these tables to contain complete information about the analyses presented in the main text of the manuscript. Instead, we have instead clearly stated the rationale behind prediction 3c in the methods section (L576-580).

Line 122: It has not been mentioned before that the soils have been treated. Please add this information to the introduction.

We now state that there was a soil treatment in the last paragraph of the introduction: “We manipulated and measured plant-herbivore and plant-soil biota interactions in 160 mesocosm grassland communities, ...” (L133). Furthermore, following comments from other reviewers, we have added a short introductory paragraph and hypothesis about potential indirect effects of soil biota on herbivores (L110-130), as well as expanding the discussion of the results of the soil treatment (L378-391).

Line 128-130: This sentence is unclear. Do you mean: “High total herbivore biomass could amount to proportionally low herbivore biomass on exotics, if exotic plants had higher biomass”?

We have edited this sentence for clarity: “Although high herbivore biomass could amount to proportionally low herbivore biomass for plants with high biomass (i.e., promoting enemy release), ...”.

Lines 137-139: In the data collection section, it is not described that you assessed which species were reproducing. How did you do that?

We have added how we assessed whether or not a species had reproduced within the mesocosms: “whether or not the herbivore species produced multiple generations in the mesocosms, as indicated by the presence of younger life stages or larger abundance than the initial introduction”. (L197-199 in Appendix S3).

Line 156: Was damage on average almost double?

Yes, we have clarified this in the text.

Fig. 2 G and H should be removed from Fig. 2, and turned into a Fig. 5, since they are treated only after Fig. 3 and 4

Agreed. We have adjusted the figures as suggested.

Line 164-165: See comments above: if here, it is about biomass produced and not biomass removed by herbivores, this should be made much clearer. I suggest not to refer to this as ‘impact on biomass’, but maybe rather ‘reduction in biomass production in mesocosms with herbivores’

We have changed the phrasing as suggested here and other places through the manuscript, which has helped improve its clarity.

Line 166: I suggest reformulating to “Exotic plants produced/had 31% less biomass in ...”. Otherwise this effect can be confused with the effects of herbivores removing biomass by chewing.

Thank you for pointing this out. We have changed as suggested (L185).

Fig. 3B: I suggest removing this fig. from the main text, since it is not related to any of the research questions.

We have moved this figure to the appendices as suggested.

Line 208: This sounds like you checked for the growth -rates – please reformulate (or present respective results)

We have removed the term “growth rate”.

Lines 238: Please discuss somewhere what this reduction in biomass may mean: I guess it is not only biomass lost by chewing, but also reduced growth? What about belowground effects? See also my comments above

Yes, the reduction in biomass in added-herbivore compared to reduced-herbivore mesocosms is considered to be a measure of the ‘net’ impact of herbivores on plants. However, we cannot reliably partition the biomass reduction among the different mechanisms mentioned by the reviewer (e.g., chewing damage, sucking insect damage, or reduced growth). We have expanded the discussion to incorporate this information (L273-286).

We have now also repeated the analysis of the impact of the herbivore addition/reduction treatment on plant biomass, but analysing above- and belowground biomass separately (the original manuscript only examined total biomass). The findings were similar to those of total biomass, but revealed subtle differences in the impacts of herbivores on different biomass partitions. For example, total biomass did not differ between native and exotic plants, regardless of the herbivore treatment, whereas when herbivores were absent, exotic plants produced 7 times more belowground biomass than natives. These findings add another layer of depth to our study and therefore we have now added all of the relevant information to the results (L191-210), discussion (L278-282), and appendices (Fig. S2.1, 2.2, 2.3).

Line 493: In the data collection section, it is not described that you assessed which species were reproducing. How did you do that?

We have now added this information (see similar comment above for details).

Reviewer 3

First, I would like to apologize for reviewing this again – the change of title, combined with the fact that this is my busiest teaching time led me to accept the review quickly without realizing I had reviewed it before – I would have stepped back had I realized (not really fair for you to come up against the same reviewer more than once!). Having said that, I think the revisions you have made are fantastic. This is now a beautifully written paper that presents the results from an extremely impressive experiment on a very important topic. I think it will be of high interest to many ecologists and land managers worldwide.

Thank you for the positive review and for the kind words about our manuscript. The constructive comments from your previous review were instrumental in helping to improve the manuscript and we are very pleased to read that you appreciated our revisions.

You have done an amazing job of responding to all my original comments. I have only two remaining suggestions:

1. You have discussed the fact that differences between native and introduced species in traits like N-fixing ability and SLA might affect your findings (thank you!). However, the reader is still left wondering whether the higher rates of herbivory on exotic species might be attributable to these factors, or if the species' provenance is actually important in its own right (which is an important question for accurate interpretation of your findings). I don't think this would be a difficult analysis (one might compare herbivory ~ traits with herbivory ~ traits * provenance and ask whether the model that included provenance explained significantly more of the variation than did the model with traits alone).

We agree with the reviewer that quantifying how herbivory varies with plant traits and provenance has the potential to improve mechanistic understanding of the plant-herbivore interactions observed in our mesocosms. We have conducted the suggested analyses and included them as Appendix S3 and in the discussion section. We included these analyses as an appendix rather than in the main text for several key reasons:

- 1) The trait-based analyses were not a part of the original scope and research questions of the paper and these questions do not fit neatly with our research objectives.
- 2) The experiment was not designed to examine the influence of plant and herbivore traits on plant-herbivore interactions. Therefore, in some instances we can only have limited confidence in inferences based on the trait-based analyses. For example, there was only one native legume species included in the mesocosms, meaning that comparing herbivory of native and exotic legumes would not be statistically defensible enough to derive robust conclusions.
- 3) Because of the sheer number of different response variables and plant and herbivore traits that were measured, the addition of these data to the main text would rapidly blow out the length of the paper, but with little change to the overall conclusions or message (e.g., no changes would be made to the abstract based on the trait-based analyses).
 - Response variables: herbivore presence, herbivore biomass, herbivore:plant biomass ratio, normalized degree, potential for apparent competition
 - Plant and herbivore traits: functional group, nitrogen fixer status, mycorrhizal fungi association, specific leaf area, total plant biomass, reproduction status in mesocosms
- 4) No clear story emerged from the results of the trait-based analyses. The importance of various traits and their interactions with plant provenance varied considerably among response variables, with no clear patterns emerging. Presenting and interpreting these inconsistent and scattershot results would result in a much lengthier and less focused paper, without adding to the overall story or helping to explain the results from tests of our original research questions. Regardless, we still include the results as an appendix and discuss the potential importance of traits in the discussion (L339-346).

2. I wondered why you didn't make more of the soil treatment in your manuscript – it is pretty impressive and unusual for a study to simultaneously consider above-ground and below-ground interactions. It may be a null result, but it is still interesting and important. We have now expanded the manuscript to further elaborate on the context and results of the soil treatment. Specifically, we have now included: 1) a mention of the treatment and results in the abstract; 2) a short introductory paragraph that provides background on the potential indirect impacts of soil biota on herbivores (L110-130); 3) a research question and specific

predictions that focus on the soil treatment (L145-147, Table 1, Figure 1); 4) a separate section for the brief results relating to the soil treatment (“The plant-soil feedback soil treatment had little influence on any of the response variables, except for moderating the relationship between proportion of exotic plants and total and belowground plant biomass as described above (all slopes $P > 0.128$). The soil treatment and its interactions did not influence herbivore presence, richness, biomass, chewing and scraping damage, or the herbivore:plant biomass ratio for either individual plants or mesocosms (all main effects and interactions: $P > 0.091$; Tables S2.1 to S2.9).”); and 5) a paragraph in the discussion where we interpret these results (L378-391). We felt that this addition was worth the extra space because soil biota could operate similarly to insect herbivores, as mediators of plant invasion, and therefore analogous hypotheses could be proposed and tested as part of our broader focal questions.